# A Comprehensive Review on the Synthesis and Versatile Applications of Biologically Active Pyridone-Based Disperse Dyes

**DOI:** 10.3390/ijerph17134714

**Published:** 2020-06-30

**Authors:** Alya M. Al-Etaibi, Morsy Ahmed El-Apasery

**Affiliations:** 1Natural Science Department, College of Health Science, Public Authority for Applied Education and Training, Fayha 72853, Kuwait; 2Dyeing, Printing and Textile Auxiliaries Department, Textile Industries Research Division, National Research Centre, Cairo 12622, Egypt; elapaserym@yahoo.com

**Keywords:** disperse dyes, self-cleaning, ultraviolet protection factor, pyridines, biological activities

## Abstract

This review summarizes our contributions during last decade on the synthesis of arylazopyridones that may be used as disperse dyes for hydrophobic fabrics utilizing an environmentally benign high temperature dyeing method. The review also discusses the advantages of select disperse dyes based on pyridone moieties as antioxidant, antimicrobial and anticancer agents.

## 1. Introduction

The instincts behind love of color differ from place to place and from one culture to another. Beautiful colors reflect the aesthetic sense of human beings that differs between individuals. Synthetic dyes are a category of high-quality natural materials, basically utilized for dyeing or printing materials that stick to the fabrics by forming a covalent bond during the application procedure. The utilization of natural dyes in textiles has largely been replaced by synthetic dyes because these give a wide assortment of reproducible hues and shades [1]. Heterogeneous amines have been widely used in preparing disperse dyes. These dyes show excellent affinity for polyester fabrics. The disperse dyes are organic colors that are less soluble in water and are widely applied to hydrophobic fibers such as polyester fibers where the dispersed dyes literally dissolve and produce the desired color [2]. The main reason for the development of dispersed dyes is the large increase in the global production of polyester fibers over the past two decades, as a large amount of these dyes is used mainly to dye polyester fabrics [3,4,5]. Some significant heterocyclic diazo components, for example, isothiazoles, thiazoles, thiadiazoles, 4-oxoquinazolines and thiophenes, give generally excellent disperse dyes with good fastness properties. In this review we focus on our contributions to the synthesis and applications of azo disperse dyes based on pyridone derivatives. The target molecules are biologically active and are used as antioxidants and anticancer agents. Environmentally benign uses of these dyes during the last decade are also described.

## 2. Chemistry of Pyridone Disperse Dyes

It is of value to mention here that pyridione derivatives have largely supplanted anthraquinones as disperse dyes with desirable properties. Among these dyes compounds **1–8** are commercially accessible (Figure 1) [6,7,8]. Azo pyridone dyes are significant pyridone derivatives that have to a great extent replaced disperse dyes based on pyrazolones due to their excellent hues.

Disperse dyes based on the pyridone moiety can be formed from *β*-diketones and different diazonium salts, followed by condensation with cyanoacetamide as a first reaction route. Mijin et al. [9,10] reported the synthesis of 5-(substituted phenylazo)-6-hydroxy-4-methyl-3-cyano-2-pyridones, i.e., the known azo pyridone disperse dyes **11a,b** via the reaction of ethyl acetoacetate (**9**) with aryldiazonium salts to give the corresponding ethyl 3-oxo-2-(substituted phenylazo)-butanoates **10** which are then condensed with cyanoacetamide in the presence of potassium hydroxide as a catalyst by either a conventional method with lower reaction yields or by a microwave heating method (Scheme 1).

A second reaction route for the synthesis of these azo dyes involves the reaction of pyridone as a coupling component and various diazonium salts. Lately, numerous researchers [10,11] have developed effective syntheses of new substituted aryl and heteroarylazoazines as potential antimicrobial dyes starting from pyridones **13a–s** (Scheme 2).

In this manner, a few researchers developed green syntheses of these pyridone derivatives using microwave [10] or ultrasound irradiation [11].

Pyridone derivatives have found wide application in the synthesis of azo dyes, particularly as disperse dyes [12,13,14,15,16,17,18,19,20]. An example of this the work of Balalaie et al. [21,22,23] who have announced an effective three component condensation of alkyl cyanoacetates, primary amines, and *β*-ketoesters affording more significant yields of **13a–h**. These compounds have two tautomeric structures, and in solution there is a quick equilibration between them. Pyridones **13a–s** could be promptly coupled with heteroaromatic diazonium salts producing the comparable heteroaromatic azopyridones dyes (Scheme 2).

Ashkar et al. [24], Sakoma et al. [25] El-Apasery et al. [26], and Al-Etaibi et al. [27] have evaluated 3-(*p*-substituted phenylazo)-6-pyridone dyes **14a–d**, **14e–n**, **14o,p** and **14q–w**, respectively, as potential disperse dyes that exist in the hydrazone tautomeric form as shown in Figure 2 for the dyes **14q**, **14r** and **14w** and applied them on polyester fabrics, to analyze the impact of substituents on the shade of the synthesized dyes.

Ashkar et al. [24], Sakoma et al. [25], and El-Apasery et al. [26] reported that the exhaustion of the colors was generally excellent on polyester fabrics, with acceptable wash and light fastness properties. These dyes are essential for their excellent affinity for fabrics. Other remarkable attributes of these dyes are that they give deep and bright hues.

## 3. Absorption Spectra Characteristics

It should be noted here that the λ_max_ values will be determined in general by the strength of the electronic force in the benzenoid framework. Since the electronic transitions of these prepared dispersed dyes involve a general movement of the electron density from the donor group towards the azo group, the best effect with respect to a longer wavelength is achieved by placing the substituent in a position *ortho* or *para* to the azo group for successful conjugation. The maximum absorption of **14q–y** colors was estimated in dimethylformamide (DMF) and listed in Table 1, sorted from 410 to 468 nm.

The colors shifts result from changes in substituents on the diazonium part of these dyes. In this way, the introduction of an electron-donating substituent on the arylhydrazono moiety brings about a critical bathochromic shift and the greater the strength of the electron-donating substituent, the more noteworthy the bathochromic move. Along these lines, dyes **14r**, **14s**, **14t** and **14x** (λ_max_ 468, 438, 464 and 445 nm) showed critical bathochromic shifts compared to dye **14q** (***λ*_max_** 410 nm) (Δ***λ*_max_** = 58, 28 and 54, 35 nm), attributable to methyl, methoxy, hydroxyl and *o*-CH**_3_** groups, respectively. The observed bathochromism upon introduction of electron donating methyl, methoxy, and hydroxy groups is in agreement with the theory. The increased electron densities added to the delocalization are because of the electron donating methyl, methoxy, and hydroxy groups are the reason for the observed bathochromic shifts, whereas with the addition of electron-withdrawing groups such as *p*-Br and *p*-Cl at the *para*-position of the arylhydrazono portion, a small bathochromic move could be observed, for example in dyes **14u**, and **14v** (λ_max_ 417 and 416 nm), can be clearly observed by showing the contrast between their bathochromic shifts compared to dye **14q** (Δλ_max_ = 7 and 8) [28].

## 4. Dyeing Characteristics

Compounds **14q–y** were utilized for dyeing polyester textures through a high-pressure high- temperature dyeing technique. After the dyeing process, the dyed fabrics ranged in color from yellow to dark orange. The dyeing properties of polyester fabrics were then evaluated with respect to their fastness properties. The K/S estimates recorded in Table 1 show that the dyes **14q–w** showed very high affinity for polyester fabrics and that all K/S values were generally good. The results recorded in Table 1 also show that the introduction of electron withdrawal groups in the benzene ring improved the lightness and brightness; conversely, the incorporating of electron-donating groups reduced the brightness, so the dyes **14u–w** were lighter and more brilliant than the dyes **14r–t**.

The washing and perspiration fastness evaluation results, listed in Table 2, were generally excellent. When the reduction clearing was overlooked, worse staining was observed. Since these dyes are naturally hydrophobic, high washing and perspiration fastness are expected. The light fastness of the dyes **14q–w** on polyester fabrics showed moderate fastness. Light fastness is mainly affected by the nature of the substituents on the diazonium portion. The incorporation of electron-withdrawing substituents (bromine, chlorine, or nitro) improves the light fastness as it reaches values of 3–4, 3–4, and 5, respectively. One feature worth mention here is the light fastness obtained by the dye **14w** which contains nitro group in the diazonium part. The nitro group increases the polarity of the dye, which may link it more strongly to the fabric and it opens an additional path of energy dissipation after absorption of light which reduces the chances of photofading [28]. 

## 5. Antimicrobial Activities

The inhibition zone diameter data for the arylhydrazopyridone disperse dyes **14q–w** against *Bacillus subtilis* and *Staphylococcus aureus* (Gram positive bacteria), *Escherichia coli* and *Pseudomonas aeruginosa* (Gram negative bacteria) and *Candida albicans* (yeast) revealed that all tested dyes have potent positive antibacterial activities against four of the five tested microorganisms. Disperse dye **14q** indicated a considerable cytolytic impact following five days of incubation [28]. After confirming with certainty that this type of disperse dyes has biological activity and that studies in this field are promising and further investigations are warranted, we synthesized the new arylhydrazono-1.4-diethyl-2,6-dioxo-tetrahydropyridine-3-carbonitrile dye **14x** and the aryl- hydrazono-1-butyl-4-ethyl-2,6-dioxotetrahydropyridine-3-carbonitrile dye **14y** in an innovative and easy way by reacting pyridones **13r** or **13s** with aryldiazonium salts. Yields were excellent and these dyes demonstrated acceptable antimicrobial activities [29].

Colors **14x**, and **14y** were utilized for dyeing polyester fabric with 2% shading, using a commercial carrier HC and Tanavol EP 20017 (supplied by TANATEX Chemicals B.V., Ede, Gelderland, The Netherlands) as an environmentally friendly carrier at dyeing temperatures of 100 °C. The data in Table 3 reveals that achieving the dyeing procedure using the eco-friendly carrier yielded results in a better color depth K/S (4.74 and 3.46) than the non-eco carrier whose results were 4.04 and 3.01 [30].

## 6. In Vitro Cytotoxicity Screening

We tested the anticancer activity of the dyes **14x** and **14y** against the HepG-2, HCT-116, A-549 and MCF-7cell lines. Various concentrations of disperse dye **14x** and disperse dye **14y** were utilized for evaluating the IC_50_ (µg/mL). Figure 3, Figure 4, Figure 5, Figure 6, Figure 7, Figure 8, Figure 9 and Figure 10 show that dye **14x** had strong activity 23.4 (HepG-2), 62.2 (HCT-116), 28 (A-549) and 53.6 (MCF-7) µg/mL, respectively, while dye **14y** revealed weak action 196 (HepG-2), 482 (HCT-116), 242 (A-549) and 456 (MCF-7) µg/mL, respectively [30].

## 7. Antioxidant Activities

We evaluated the antioxidant properties of the two dispersant dyes prepared by us in an easy and effective way in vitro by their 2,2-diphenyl-1-picrylhydrazyl (DPPH) free radicals scavenging activity. Figure 11 and Figure 12 demonstrate the good antioxidant activity of the dispersant dye **14x**, with IC_50_ of 64.5 (more than ascorbic acid used as a standard, with an IC_50_ of 14.2), while the disperse dye **14y** had weak antioxidant activity (IC_50_ 191.6) [30].

## 8. High Temperature Dyeing as Environmentally Benign Method

During our study, we have taken into account the ability of disperse dyes to preserve the environment by reducing dye residues by coloring polyester fabrics with the dyes **14x** and **14y** using a high pressure and temperature dyeing strategy. The degree of dye exhaustion in dyebaths has been compared to the low-temperature dyeing technique as dyebath wastes directly and negatively affect the environment. It should be noted that the degree of absorption of the dye by the high-pressure high-temperature (HT) dyeing technique for all polyester fabrics is completely inconsistent with the low-temperature dyeing process, and therefore (HT) expanded and deepened the color strength K/S of the dyes examined, evaluated at 309 and 265% (Table 4). Thus, this gives clear evidence that the measurement of the color present in the dye resulting from the use of the high-pressure high-temperature (HT) dyeing technique is virtually non-existent and ineffective, therefore it is clearly, contaminated, and completely positive for the environment as these remaining colors in the dyebaths are typically dumped into the environment as waste, harming the environment [31].

Dyes **14x** and **14y** were utilized for coloring polyester fabrics by means of the high temperature coloring strategy, Table 4 reveals their excellent perspiration and washing fastness properties and fastness to light which was not as good for the untreated polyester colored fabrics [31]. After we evaluated the pyridone-based disperse dyes and their resistance to bacteria, we also evaluated and tested untreated polyester fabrics dyed with dyes **14x** and **14y** using the high-temperature and high-pressure dyeing method for their inhibitory effect on the development of six different microorganisms. The antimicrobial screening results showed that untreated fabrics dyed with dye **14x** had no antibacterial properties against the tested pathogenic bacterial strains, while untreated polyester fabrics dyed with dye **14y** had good antibacterial properties against *Staphylococcus sciuri* (13 mm), strong antibacterial activity against *Pseudomonas aeruginosa* (24 mm), and very strong antibacterial properties against *Escherichia coli* (39 mm) [31].

## 9. The Multifunctional Performance that nano TiO_2_ Imparts to Polyester Fabric

In an attempt to acquire fabric dyed by the disperse dyes, the post-treated polyester fabrics dyed with disperse dyes **14x** and **14y** was subjected to a two-step hot process after they were completely immersed in a solution of TiO_2_ NPs nanoparticles at 80 °C then we performed a curing process at 140 °C. Here, we can say that the treated fabrics are recognized to provide ideal UV protection factors ranging from 34.9 and 283.6 (Table 5) [31].

The dyed fabrics treated with TiO_2_ NPs additionally exhibited strongly improved light fastness properties, where it became 6 for dye **14x** and also 5–6 for dye **14y**. Likewise, the possible uses of these fabrics treated with titanium nanoparticles have been studied, for example, anti-fungal and self-cleaning activities. Interestingly, the polyester dyed fabrics with treated TiO_2_ NPs and disperse dye **14x** had strong antifungal properties against *Aspergillus flavus* and *Penicillium chrysogenum*. This is predictable in light of published data [32] where the treatment of polyester fabrics with inorganic NPs oxides may confer upon these fabrics an antimicrobial function (Table 6). 

## 10. Conclusions

The large number of uses of disperse dyes have caught the attention of many researchers who have emphasized their importance in the last decade. Our contributions in the field of applied chemistry have been presented through this review article which aims to provide a comprehensive overview of the new disperse dyes that we have synthesized in our laboratories *via* an environmentally safe route to reduce environmental pollution and demonstrate the characteristics and multiple uses of these new dyes. Since manufacturing of the first disperse dye began in 1923, the prevalence of disperse dyes has made them today the second largest sector in the dye industry. The dye industry faces the challenge of manufacturing high-quality disperse dyes, and this is one of the driving factors for its research and development studies. Nowadays a wide range of dyes is being created on the basis of the potential end-uses such as dyes for sportswear, automotive fabrics and transfer printing. These days efforts are being made to invent dyes for dispersant-free dyeing to reduce the pollution caused by dyeing polyester fabrics with these dyes, and this is what we indicate in this review article about the importance of disperse dyes and their direct correlation with the environment.

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
