# Peer review of "A Comprehensive Review on the Synthesis and Versatile Applications of Biologically Active Pyridone-Based Disperse Dyes"

_ijerph, 2020, doi:10.3390/ijerph17134714_

Round 1

Reviewer 1 Report

This manuscript provides a comprehensive review of the synthesis and applications of the pyridine disperse dyes for hydrophobic fabrics. The review is complete, and the ideas are clear, however, some minor revisions are needed before the acceptance by IJERPH:

  1. In the title , the authors used the phrase “our contributions on…”, while it is fine to only review the authors’ own work, it is not necessary to put it into the title which is already long.
  2. In page 3, related to Scheme 2, the author mentioned “numerous researchers have developed effective synthesis of…”, where some references should be provided.
  3. The abbreviations in the manuscript need to be interpreted when they appear for the first time, such as “DPPH”(2,2-diphenyl-1-picrylhydrazyl) and “UPF”(ultraviolet protection factor).

Reviewer 2 Report

Specific comments:

  1. In page 6, ‘methoxy and hydroxyl’ should be un italicized.
  2. Should provide a list of the tested bacterial microorganisms in the antimicrobial activity section.
  3. Dye 14x had strong activity 23.4 ± 1.2 and 62.2 ± 4.1 against the which cell lines? A similar problem with dye 14y also.
  4. Should add units of IC50 values.
  5. In conclusion, it should include the main rational and consequences of dispersed dyes.

Reviewer 3 Report

The paper describes various work by the authors regarding the synthesis and evaluation of various dyes. En general the work is interesting as it gives a good overview by bringing the various works together.

In general the paper is ok to publish but I believe needs some serious editing before it is fit for publication.

(1) The english needs to be checked as some constructions sound somewhat unatural.

When the authors used the word "clubbed" do they mean "coupled"? Or does this have some special meaning in the context of this field that I am unaware of??????

(2) It seems that in assembling the article (from different previous works) the authors have not changed the drawing settings. Figure 1 , Scheme 1 and Scheme 2 all have different drawing settings and sizes!

Also the arrangement of the central nitrogens in Figure 1 are drawn in a linear fashion which is incorrect  - they are shown correctly in scheme 2! Also the structure numbers need to be in bold.

The same happens for the tables which all have different text sizes.

These erros of formatting give a poor look to the article and need correcting (choose one or use the reccomended template fro the journal!!!!)

With regards to the conclusion - I felt it could do with rewriting as it was partially incorrect and somewhat overstated as well as somewhat grammatically dubious.

Having looked up the definition of "social economy" I believe it does not apply here. Environmentally friendly o safer to use maybe but I think anything beyond that is exaggerated.

The authors state "...the most impressive and sophisticated industrial resources" when referring to these compounds. While I acknowledge the benefits I feel this is a bit much for a type of compound that has been in common use since the start of organic chemistry as a scientific study.

Please rewrite the conclusion. Maybe the authors could give some more specific examples of the potential for these compounds rather than just stating things in very general terms. As well any outlooks for the future potential of these compounds etc etc.

Once the errors are corrected I would be happy to accept the paper for publication.
